# Applications of Biomaterials in 3D Cell Culture and Contributions of 3D Cell Culture to Drug Development and Basic Biomedical Research

**DOI:** 10.3390/ijms22052491

**Published:** 2021-03-02

**Authors:** Yujin Park, Kang Moo Huh, Sun-Woong Kang

**Affiliations:** 1Department of Polymer Science and Engineering & Chemical Engineering and Applied Chemistry, Chungnam National University, Daejeon 34134, Korea; yujin.park@kitox.re.kr; 2Predictive Model Research Center, Korea Institute of Toxicology, Daejeon 34114, Korea; 3Human and Environmental Toxicology Program, University of Science and Technology, Daejeon 34114, Korea

**Keywords:** 3D cell culture, biomaterials, drug screening, alternative model

## Abstract

The process of evaluating the efficacy and toxicity of drugs is important in the production of new drugs to treat diseases. Testing in humans is the most accurate method, but there are technical and ethical limitations. To overcome these limitations, various models have been developed in which responses to various external stimuli can be observed to help guide future trials. In particular, three-dimensional (3D) cell culture has a great advantage in simulating the physical and biological functions of tissues in the human body. This article reviews the biomaterials currently used to improve cellular functions in 3D culture and the contributions of 3D culture to cancer research, stem cell culture and drug and toxicity screening.

## 1. Introduction

The human body consists of highly sophisticated biological systems [1]. Cells form tissues in various combinations and patterns, tissues form organs with different types of tissues, and organs are organically connected to maintain the human body [2]. For a long time, two-dimensional (2D) cell culturing has been carried out on widely available flat plastic dishes to mimic the complex human body [3,4]. However, in a 2D culture system, the cells spread on the flat and hard surfaces and proliferate unnaturally. There is a difference in the cellular morphology, functions, and overall behavior compared to those in the natural environment [4]. In addition, the phenotype of the cell is not accurately reproduced in 2D culture. Indeed, chondrocytes grown in vitro to a large number gradually lose their differentiated phenotype, which is indicated by the loss of synthesis of type II collagen during 2D cell culture [5,6,7]. Similarly, 2D cultured primary human hepatocytes (PHHs) show rapid declines in critical phenotypic functions, such as cytochrome P-450 (CYP450) enzyme activities, insulin responsiveness, and expression of the master liver transcription factor hepatocyte nuclear factor 4α within hours to days [8]. Moreover, the transwell culture system, a kind of layered 2D culture system, was introduced as a co-culture system to simulate the in vivo environment, but this method has limitations in maintaining or improving cellular function over a long time [9,10]. Thus, cell culturing should be adapted to better reflect the natural environment.

Cells in the natural environment are embedded in the extracellular matrix (ECM), forming a complex three-dimensional (3D) structure [11]. The ECM plays the role of regulating cell-to-cell interactions, cell adhesion, differentiation, and growth [12,13,14]. Therefore, an understanding of ECM composition and structure is critical for the development of novel 3D cultures for predicting biological mechanisms and therapeutic effects. Mounting evidence has shown that physiologically more relevant factors can be revealed by imitation of the components and structure of the ECM in the natural environment [13,15,16]. In particular, cells cultured in a 3D microenvironment with ECM components showed realistic morphology and expressed several genes that failed to be expressed in a 2D culture [5,6,7]. Moreover, these cells synthesized ECM as they do in vivo for regeneration [11,12,13,14]. Thus, 3D cell culturing requires the use of biomaterials with a high level of similarity with the ECM for the enhancement of cellular functions.

A number of 3D culture systems are already available (Table 1). Biomaterials are available for 3D cell culture to improve the efficiency of culture and cell functions in various forms, including hydrogels, solid scaffolds, decellularized native tissue, and ultra-low attachment (ULA) surface (Figure 1). Knowledge of 3D culture methods has significantly increased, which has resulted in the development of numerous applications. Thus, this review addresses the applications of biomaterials in 3D cell culture and the contribution of 3D cell culture to cancer research, stem cell research, and drug and toxicity screening.

## 2. Applications of Biomaterials in 3D Cell Culture

### 2.1. Hydrogels

Hydrogels have 3D structure, hydrophilic property, and polymeric networks capable of absorbing large amounts of water or biological fluids [32]. These hydrogels can mimic soft and wet environments similar to ECM of native tissues and promote the transportation of O_2_, nutrients, waste and soluble factors [33]. Therefore, they have received much attention in 3D cell culture [6,14]. Hydrogels are categorized as either synthetic or natural (Table 2). As the name suggests, synthetic hydrogels are composed of unnatural molecules such as poly vinyl alcohol (PVA), poly-2-hydroxyethyl methacrylate (pHEMA), and poly ethylene glycol (PEG). These materials can provide mechanical support for various types of cells [14]. However, they are biologically inert. In addition, they lack endogenous factors essential for cell behavior and act mainly as a template to permit cell function. Thus, synthetic hydrogels need modification with suitable biological components to promote signals of cellular function. On the other hand, synthetic hydrogels such as PEG represent a very good candidate for the encapsulation of various bioactive factors, drugs, and chemicals to avoid complicating systemic factors derived from hydrogels for a more controlled comparison of encapsulated materials [34]. Recently, it has been reported that the combination of arginine-glycine-aspartic acid (RGD) groups or alginate-PEG hydrogel improved the spread and proliferation of fibroblasts and enhanced the osteogenic differentiation of mesenchymal stem cells (MSCs) for 3D cell culture [35]. Similarly, PEG hydrogels were used to culture and expand a variety of neural and glial cell types simply by altering the material properties of the hydrogel [36].

Natural hydrogels are made up of natural substances such as collagen, alginate, hyaluronic acid and many more that promote several cellular functions with a range of endogenous factors present, which can benefit the viability, proliferation, and differentiation of many cell types [47,48,49,50,51,52,53]. However, due to the complexity and undefined nature of these hydrogels, it is difficult to very accurately determine which signals promote cellular function. Collagen is one of the abundantly present proteins in the ECM [54]. These compounds possess native tissue-like properties and characteristics. Thus, collagen can be used to create gels for 3D cell culture [48,55]. Collagen can be used in culturing various cell types to improve cell growth, adhesion, and differentiation. Cells can proliferate and form tissue-like structures within the collagen matrix [56]. Collagen plays an important role in maintaining the chondrocyte phenotype and supporting chondrogenesis, both in vitro and in vivo [57,58]. Previous studies have shown that type I collagen promotes the proliferation of chondrocytes and that type II collagen supports the chondrogenic differentiation of MSCs [59,60]. The results from this study suggest that there is clinical value in the cartilage repair capabilities of Col I/II hydrogel with encapsulated MSCs [61,62]. Nonetheless, the long-term performance of pure type I collagen may be compromised by significant shrinkage and weak mechanical properties [63,64]. To control these problems in collagen hydrogels, one possibility is to introduce additional molecular bonds between the collagen fibrils via different chemical cross-linkers. Lotz et al. aimed to improve the long-term stability and mechanical properties of collagen hydrogels by using the nontoxic chemical cross-linker four-armed succinimidyl glutarate polyethylene glycol (PEG-SG) to obviate negative impacts on cell viability. The hydrogels showed increased mechanical stability and compression E-modulus compared with pure collagen. This could indicate a more sterically rigid molecular network, rendering human dermal fibroblasts, and human epidermal keratinocytes unable to contract the hydrogel. This leads to a reproducible generation of full-thickness skin equivalents for in vitro testing or clinical application [65]. Incorporation of other materials into polymeric hydrogels can also be a suitable option to overcome these problems and improve the biological performance of the hydrogels. Sun et al. observed that collagen–chitosan could promote axonal regeneration and neurological recovery compared with collagen–chitosan hydrogels fabricated by traditional technology. In addition, it was demonstrated that 3D printing of collagen–chitosan decreased the formation of scars and cavities, and improved the regeneration of nerve fibers as well as functional recovery in rats [66,67]. Ying et al. fabricated a porous structure of this collagen–hyaluronic acid (HA) hydrogel that contributed to water retention, gas exchange, nutrition penetration, and cell dwelling [68]. In addition, these materials are suitable to study tissue reconstruction by seeding co-cultures of fibroblasts and endothelial cells within the collagen matrix [69]. In this system, fibroblasts form connective tissue, and endothelial cells produce angiogenic growth factors and vasculature.

Hyaluronic acid (HA) is distributed in many tissues, such as skin and cartilage [70]. HA can be obtained not only from animal tissues, but also via microbial fermentation in *Escherichia coli* to produce animal-free HA [71]. The role of HA in tissue is to promote cellular survival, migration, angiogenesis, and differentiation by transduction of intracellular signals [72,73,74]. In addition, the higher content of HA present in the cancer microenvironment promotes tumor progression and resistance to anticancer drugs [75,76]. Tumor cells showed decreased adhesion to the surface of HA. These properties promote the production of tumor spheroids and mimic cell HA signaling in the tumor microenvironment for anticancer drug screening purposes. Ahrens et al. reported that HA promotes the growth rate of melanoma cells by enhancing the secretion of basic fibroblast growth factor (bFGF) [77]. Other researchers have reported that when cells are 3D cultured in the presence of HA, the activity of multidrug resistance proteins is enhanced and therapeutic effectiveness is reduced compared to the 2D cultured cells [78]. Another interesting application of HA hydrogel is to improve the efficacy of 3D cell culture by mixing these materials with various substances. Lou et al. reported that HA-collagen hydrogels promoted cell spreading, fiber remodeling, and focal adhesion in 3D cell culture [49]. Häckel and coworkers demonstrated that human nucleus pulposus cells cultured in fibrin-HA hydrogels showed an increase in collagen type II and carbonic anhydrase XII gene expression [79]. Furthermore, Lee et al. also reported that chitosan/HA blend hydrogels exhibited enhanced physical stability, mechanical properties, cell binding affinity, and tissue compatibility [80]. Recently, HA combined with alginate and fibrin has been used as a bioink for 3D bioprinting of peripheral nerve tissue regeneration [81]. Finally, acetylated HA (AcHA) was used to enhance the mechanical strength of the thermogel via simple blending of modified glycol chitosan. The blended gel showed not only good cell binding affinity in vitro and biocompatibility in vivo, but also more effective cartilage formation than that of the original hydrogel [80].

Alginate is derived from the cells of brown algae, and its monomers have the ability to cross-link to form hydrogels [50]. Normally, alginate does not interact directly with mammalian cells and is not degradable [82]. Thus, when hydrogels exhibiting minimal degradation are desired, alginate is selected for these studies. In addition, cell adhesion can be improved via covalent coupling of the RGD cell adhesion peptide to the alginate chains [83]. A previous study reported a material approach to tune the rate of stress relaxation of hydrogels for 3D culture, independent of the hydrogel’s initial elastic modulus, cell adhesion ligand density, and degradation. The influence of substrate stress relaxation on cell spreading and proliferation was enhanced when RGD cell adhesion and ligand density was increased in gels with faster relaxation [84]. Another study reported that stem cells encapsulated in ionically crosslinked alginate hydrogels undergo predominantly adipogenic differentiation at initial moduli of 1–10 kPa and predominantly osteogenic differentiation at initial moduli of 11–30 kPa [85]. Recently, alginate hydrogels have been extensively used as bioinks to provide 3D cell growth because of their relatively higher viscosity and rapid crosslinking process after printing [86,87]. In addition, oxidized alginates showed great potential as ink for bioprinting [88]. Finally, alginate hydrogels encapsulating stem cells have been investigated for the prevention of immune rejection of transplanted cells [89,90]. Stock and coworkers reported that alginate capsules prevented infiltration of immune cells while allowing smaller molecules, such as oxygen, nutrients, glucose, and insulin to diffuse freely through the capsule [91].

### 2.2. Porous and Fibrous Scaffolds

Solid scaffold-based cell culturing is one of the older techniques used in the field of 3D cell culture [92]. In this system, scaffolds may facilitate proliferation, cell adhesion, and signaling activities between the cells. These efficacies of a scaffold are affected by the materials that make up the scaffold and its physical structures, such as exposed surface, pore size, pore distribution, and interconnectivity (Table 3). These solid scaffolds are mainly porous foams or fibrous meshes fabricated from synthetic polymers, such as poly(glycolic acid) (PGA), poly(lactic acid) (PLA), poly(lactic-co-glycolic acid) (PLGA), and polycaprolactone (PCL), and naturally derived polymers, such as collagen, hyaluronic acid, fibrin, alginate, gelatine, silk, and chitosan [93,94,95,96,97,98,99].

Porous foam-solid scaffolds have high porosity and a uniform interconnected structure. Many attempts have been made to fabricate porous foam-solid scaffolds [108,109]. Particulate leaching is a physical process that involves casting polymers around soluble beads known as porogens [100]. Solvent casting uses a polymer dissolved in an organic solvent. This solution is mixed with ceramic particles and poured into a predefined 3D mold, which is left to set. The solvent casting and particulate leaching (SCPL) method, which combines the particulate leaching method and solvent casting method, has been used to produce scaffolds for the culture of osteoblasts and osteogenic differentiation of stem cells [110]. Mouse embryonic osteoblast cells (MC3T3-E1) cultured on PLGA scaffolds made with the SCPL method showed increased alkaline phosphatase activity and expression of type I collagen [111]. Emulsion templating is one of the common methods for the fabrication of porous scaffolds [16]. Porous polymers can be generated within high internal phase emulsions. It has been reported that metabolic activity is improved when hepatocytes are cultured on PCL scaffolds containing various factors by emulsion templating [112,113]. The gas forming technique is performed by agitating the polymer and creating foam [104]. High-pressure gases such as CO_2_ can be used as the gas foaming agent, and the porosity of the scaffold can be controlled by the amount of gas dissolved in the polymer. The melt molding method uses both polymer and porogen, which are poured into a mold and heated above the polymer glass transition temperature [107,114]. Various types of cells have been successfully cultured in 3D on porous solid scaffolds [115,116,117,118]. Vascular smooth muscle cells adhered to and proliferated in engineered smooth muscle tissue on highly porous and elastic tubular scaffolds [119]. Human hepatoma cells showed higher cell infiltration in PLGA scaffold fabricated by the gas foaming method. The porous PLGA scaffold fabricated by the particulate leaching method supported cell adhesion and growth. After implantation, there was better bone and cartilage formation inside the scaffold [120,121].

Fibrous scaffolds provide a large surface area for cell growth in 3D cell culture (Table 4). These structures allow appropriate space for gas and nutrition exchange and cell infiltration. In addition, fibrous scaffolds can imitate oriented and aligned tissues, which include skeletal muscles, the central nervous system, and cardiac tissues [122,123,124,125,126]. Accordingly, the aligned fibers help control stem cell differentiation into the desired cell type [127,128,129,130]. Several natural and synthetic polymers, including collagen, gelatine, hyaluronic acid, alginate, chitosan, silk, PLA, PLGA, and others, have been used for the fabrication of fibrous scaffolds [124,129,131,132]. The fiber mesh is either knitted or woven into 3D patterns of different pore sizes [133]. However, these materials do not have sufficient mechanical and structural stability. Fiber bonding was thus developed to overcome the drawbacks of fiber mesh. Enhanced mechanical strength is provided by binding the fibers at the joints or intersections by raising temperatures above the polymer melting points or by using special solvents [134]. The electrospinning method uses an electric field generated using two electrodes (one each placed in the polymer and collector solutions) having electric charges of opposite polarity for the production of continuous fibers ranging from submicron to nanomicron diameters. This system allows cells to adhere and elongate along the fibers, which induces cell alignment and directionality in the cultures [135]. Lee et al. fabricated PLLA fibrous scaffolds using the electrospinning method, and their morphologies were controlled by the fiber collection speed. Therefore, the morphology of the designed fibrous scaffolds in this work has successfully controlled cell alignment as well as the direction of calcification [136]. For phase separations, two phases that are polymer rich and polymer poor are formed upon the addition of water, inducing phase separation [137,138]. Upon cooling below the solvent melting point followed by vacuum drying, the scaffold is obtained. This method can easily be combined with other fabrication technologies, such as particulate leaching, to design 3D structures with the desired pore morphology. Finally, nanofibers can be generated by the self-assembly of synthetic or natural molecules [139,140]. These scaffolds fabricated by self-assembly facilitated attachment and migration of hepatocytes, stem cells, and endothelial cells [141,142].

Another way to fabricate a porous scaffold is 3D printing. This method allows for easier and more detailed manufacturing than the above two methods and allows various architectures and control of mechanical stability [149]. The method can be applied to bone tissue engineering by taking advantage of its strong mechanical strength [150]. Chitosan material, which has been used as a soft scaffold in the form of a conventional hydrogel, can also be used for bone tissue engineering after reinforcement of its mechanical strength by applying this method. This can be done by incorporating a chitosan thermogel into a porous PCL scaffold [151].

### 2.3. Decellularized Native Tissue

One of the ideal scaffolds is a decellularized matrix that provides natural geometric morphology, flexibility, and mechanical strength, which is difficult to mimic perfectly with synthetic scaffolds. Recently, various approaches have been introduced to fabricate decellularized scaffolds, including perfusion of the whole organ (recommended for dense organs/tissues), application of a pressure gradient (employed for hollow tissues), use of supercritical fluid (appropriate for long-standing storage of decellularized scaffolds), and immersion and agitation (suitable for thin tissues) [152,153,154,155,156,157]. During the decellularization process, the cells are eliminated to inhibit inflammatory reactions or immediate rejection after implantation. The ECM derived from decellularized matrix provides an endogenous environment, from a biochemical and anatomical point of view, for regeneration of target organs. Recellularization is performed by direct injection of cells into the vein because of its proper vascular diameter and accessibility. This method is broadly used for blood vessel recellularization of different organs, such as the heart, lungs, and liver [158,159,160]. Another approach for recellularization is cell inoculation into mass media by allowing cells to recover through the circuit to seed scaffolds, but the efficiency of this method is lower than that of direct injection using veins [161]. In various studies, researchers investigated the effect of decellularized scaffolds on cell proliferation and construction of organs, including the liver, heart, lung, kidney, and pancreas [162,163,164,165,166,167].

### 2.4. Ultra-Low Attachment Surface

Cell culture plates can be covered with biomaterials with low cell-binding properties to prevent the cells from adhering to the surface. This method is one of the older techniques to generate self-assembled cellular structures in media for 3D cell culture. This system inhibits the attachment of cells to the surface of the culture plate, resulting in force floating of cells. Force floating improves cell-to-cell interactions, enabling multicellular aggregation. To provide attachment-resistant cell surfaces, cell culture plates or surfaces are coated with polymers that possess low cell-binding properties, such as 2-hydroxyethyl methacrylate (poly-HEMA), polyethylene glycol, chitosan, agar, and agarose [168,169,170,171,172]. These polymers allow greater cell-cell interactions rather than cell-substrate interactions, which enables spontaneous spheroid formation [173]. Cell culture plates could have flat or round surfaces. The flat surface causes the formation of irregularly sized spheroids. However, round surfaces are capable of generating single spheroids. Various types of cells have successfully formed spheroids on plates or round surfaces with low cell-binding properties [174,175,176,177,178]. In this system, cells accumulate as clusters and synthesize their own ECM. In addition, the signaling and communication between cells in spheroids were enhanced, and gap junctions were created that can facilitate the exchange of ions, small molecules and electrical currents. This method can be used for high-throughput screening [179]. Recently, Cho et al. fabricated new polymers with low cell-binding properties for spheroid generation [180]. Spontaneous spheroid formation was completed within a few days, and the size of the spheroids varied with the cell density.

## 3. Applications of Three-Dimensional Cell Culture

### 3.1. Cancer Research and Drug Screening

Three-dimensional (3D) cell culture has several applications in the field of biological science. These applications can be divided into three categories: cancer research, stem cell culture, and drug toxicity and screening. With the growing demand for effective validation in the field of oncology, there is a demand for research tools that provide clinically significant results for safety and efficacy testing of anticancer drugs. Moreover, 3D culture techniques have been able to produce 3D models, such as oncospheres, spheroids, and other tumor models that closely resemble the natural tumor microenvironment due to the proper supply of nutrients, oxygen, and intercellular interactions [181,182]. Increasing the scope of this technology could reduce the number of animals sacrificed for research. In addition, various meaningful results that could not be solved with animal models can be obtained.

Animal models and 2D culture systems have been a standard to determine anticancer drug effects on growth inhibition and apoptosis. However, the results produced from these systems showed relatively low similarity in clinical outcomes [183,184]. Owing to their close resemblance to tumor microenvironments, 3D culture systems are gradually being used to replace animal and 2D culture models for screening anticancer drugs that are in their initial stages of development. The spheroids contain proliferating cells on the surface and quiescent cells in the core due to limited penetration of nutrients and oxygen, and necrotic areas may be included in larger spheroids that resemble native solid tumors [185,186]. Several studies have shown differences in viability in 2D- and 3D-cultured cancer cells [187,188]. The cells were treated with the same concentration of tirapazamine (TPZ). Cells cultured in 2D were more resistant to treatment with TPZ and showed 72% viability. In contrast, cells cultured in 3D were more responsive to treatment with TPZ, with 40% viability. The difference was correlated with the fact that TPZ is a hypoxic activated cytotoxin, which works more effectively on cells cultured in 3D because the core of the spheroid is hypoxic due to limited oxygen diffusion. Some studies have reported that cancer cells in 3D models show increased drug resistance compared to those in 2D culture [189]. The variation in response to drugs can be explained by limitations on the mass transfer of the drugs in 3D culture systems compared to 2D cultured cells [190,191].

Cancer cells grown and maintained in 3D models show phenotypic heterogeneity [192]. This is important because cells of the same tumor group also change morphologically and functionally as gene expression, differentiation and proliferation rates change. Due to the diversity of cancer cells, it is difficult to develop drugs that accurately target tumor cells. In this regard, appropriate genetic modifications of ovarian epithelial cancer in a 3D environment have been investigated during development and progression [193,194,195]. Therefore, 3D models are used to study changes in gene expression-related tumor microenvironments. Moreover, 3D cell culture was used to confirm the presence of cancer stem cells [196]. Numerous studies have demonstrated that cancer stem cells can generate tumor-spheres by a suspension of single cells in serum-free conditions. Thus, scaffold-free methods are being used for 3D culture of cancer stem cells. In addition, 3D coculture of cancer cells and normal cells was used to confirm the specificity of drugs for cancer cells [197,198]. These studies suggest the possibility of 3D models in anticancer drug screening to more accurately represent tumor models.

### 3.2. Stem Cell Research and Drug Screening

Recently, stem cells have become an important research tool in biology, medicine and toxicology, leading to a rapidly developing new scientific field. The importance of stem cells is evidenced by the increasing number of articles published each year. This rapid development requires new means of providing high-quality, well-defined and scalable techniques for the generation of stem cells. It is essential to proliferate stem cells to be used as a tool in scientific disciplines. Embryoid bodies (EBs) are 3D aggregates generated in suspension by pluripotent stem cells (PSCs), including embryonic stem cells (ESCs) and induced pluripotent stem cells (iPSCs) [199]. EB differentiation is a common tool to generate specific cell lineages from PSCs [200,201]. Many 3D culture methods of PSCs in the form of EBs have been developed. Conventional methods for forming EBs include scaffold-free methods such as the ultra-low attachment (ULA) method, hanging drop methods, and suspension culture methods. In particular, EB formation can be more accurately controlled by the inoculation of known cell numbers within single drops suspended from the lid of a Petri dish [199,202]. While this method enables control of EB size by altering cell density per drop, the formation of hanging drops is labor intensive and does not facilitate scalable cultures. Additionally, the media cannot be readily exchanged without disturbing the EBs every 2–3 days. Recently, new methods have been developed to enable media exchange using a modified hanging drop plate [203]. In addition, these methods have also been designed to readily separate EBs within individual wells on adhesive substrates for the analysis of EBs [204,205]

One of the major efforts in the stem cell field is to develop therapeutic applications for the regeneration of diseased, dysfunctional or complex injured tissues. To that end, it is essential to secure technology for stem cell proliferation with increased cell viability and differentiation potential [206]. Various studies have been conducted to prove the convenience of direct transplantation, improved graft adherence, and increased cellular loading densities through 3D cell culture [207]. Studies have reported hydrogel encapsulation in 3D cell culture for regeneration of cartilage, bone, liver, cardiac, cortical, brain, and skin tissues [208,209,210,211,212,213,214]. This method facilitates cell growth, differentiation and migration of many cell types, such as neural stem cells, mesenchymal stem cells, adipose-derived stem cells, and PSCs. In addition, hydrogels with stem cells have demonstrated therapeutic effects in xenograft models [215].

Three-dimensional (3D) cell culture has emerged as a promising method for the generation of organoids that could serve as a model to study various disease mechanisms or the toxicity of new drugs (Figure 2). Organoids are highly heterogeneous 3D structures that exhibit typical tissue architecture. Thus, organoids can serve as a valuable developmental model to explore new drugs [216,217]. To date, several protocols have been developed for the in vitro generation of organoids for the brain, gut, retina, liver, skin, and kidney [218,219,220,221,222]. Organoid formation generally requires culturing stem cells or progenitor cells in a 3D environment [223,224]. Hydrogels such as Matrigel or collagen gel are used to provide a 3D environment. When stem cells are used for the creation of the organoid, cells are allowed to form embryoid bodies. These embryoid bodies are an example of organoids. Cardiac organoids generated from cardiomyocytes have the ability to spontaneously contract [225]. These can produce electrophysiological measurements when plated on multielectrode arrays. Hairy human skin can be generated using a unique approach of organoid formation. PSCs can be differentiated into epidermis, dermis, fat, nerves and hair buds that produce hair [226]. Several studies have shown that more realistic outcomes can be achieved to study disease mechanisms by organoids and how cells respond to drug treatments [227,228]. However, standard protocols of organoid generation have yet to be defined. Therefore, the development of organoid culture platforms for large-scale production, organoid-based high-content screening platforms, and finely controlled systems should be continued. Finally, developing organoids generated from PSCs in an organ-on-chip system is also an area to be explored. The function of various organs or tissues, such as the lung, liver, kidney, heart, and gut, can be realized by using microfluidic device technology in an organ-on-chip system [229,230]. The successful development of organ-on-chip system will be able to mimic the systemic circulation of humans and animals under in vitro conditions and will provide us with vast benefits as an alternative model for drug screening and toxicology tests.

## 4. Conclusions and Future Perspectives

Herein, we introduced 3D cell culture methods and summarized the trends in the application of this technology. The 3D cell culture technologies have already emerged as an important tool that can facilitate research in the biomedical field with more realistic results. However, 3D culture methods have many problems to overcome. First, there are no standardized culture methods. The 3D cell culture method is more difficult and more complicated than that of 2D cell culture. Thus, it has low reproducibility and requires a high level of skill. The second problem is that 3D cell culture is still inappropriate for completely simulating the intrinsic properties of cells. One example is a vascular problem. In the body, tissues are constantly supplied with nutrients and oxygen from microvessels in the tissues. Cells cannot survive beyond 150 μm from a microvessel. Thus, angiogenesis is an important consideration for successful 3D cell culture. When tissue is regenerated in vitro, the 3D tissue should be supplied with nutrients and oxygen around it until blood vessels grow into the tissue. In this case, growth factors as well as vascular endothelial cells and stem cells may be used to promote angiogenesis. However, the problem is that necrosis can occur before blood vessels are formed due to the insufficient supply of nutrients and oxygen within the 3D tissue. Finally, even if this simulation is correctly achieved, there is still a lack of research on analytical methods for verification. Most analyses now depend on visual images. Therefore, it can be difficult to clearly judge the standards and grasp the characteristics inside 3D structures. For this reason, standardization of the analysis of 3D cell culture is still difficult. Although there are still many problems to overcome in 3D cell culture, the introduction of advanced 3D cell culture technologies has attracted several researchers to shift their attention from 2D to 3D cell culture systems. Thus, 3D cell culture technology is expected to contribute greatly to future research with the potential to create excellent outputs.

## Figures and Tables

**Figure 1 ijms-22-02491-f001:**
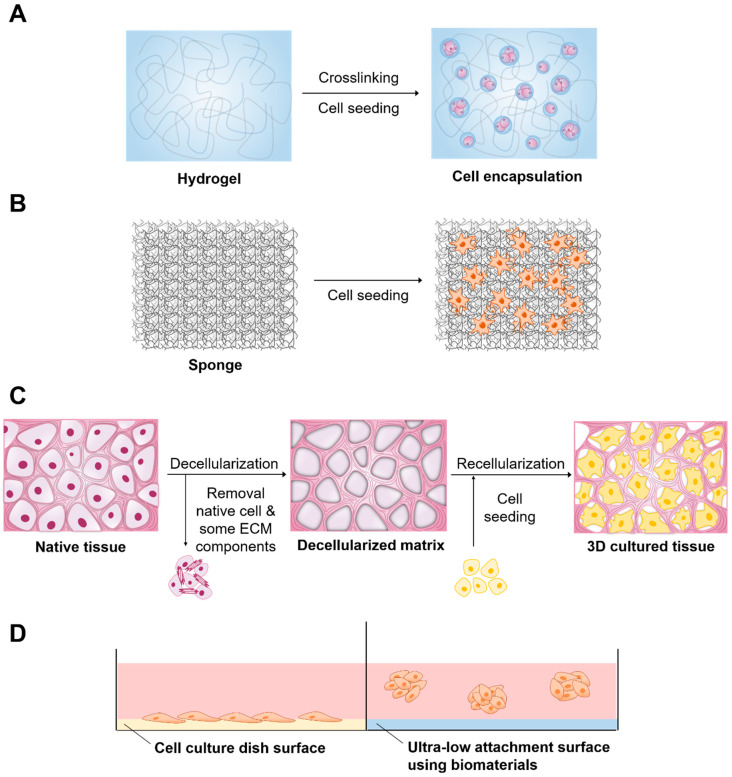
Biomaterials and related method of three-dimensional (3D) cell culture preparation. (**A**) Hydrogel, (**B**) Solid scaffold, (**C**) Decellularized native tissue (**D**) Ultra-low attachment surface.

**Figure 2 ijms-22-02491-f002:**
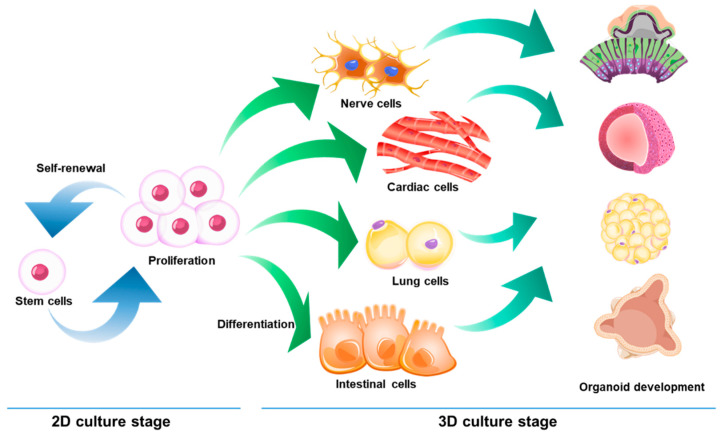
Stem cell-derived organoids.

**Table 1 ijms-22-02491-t001:** Types of biomaterials used in three-dimensional (3D) cell culture and their advantages and disadvantages.

Type	Advantage	Disadvantage	References
Hydrogel	Tissue like flexibility Easily supplies water-soluble factors to cells	Low mechanical resistance	[5,11,13,17,18]
Solid scaffold	Various materials can be used Physical strength is easily adjusted	Difficulty in homogeneous dispersion of cells	[15,16,19,20,21]
Decellularized native tissue	Provides complex biochemistry, biomechanics and 3D tissues of tissue-specific extracellular matrix (ECM)	Decrease of mechanical properties (roughness, elasticity, and tension strength) of the tissues as compared to the native group	[22,23,24,25,26]
Ultra-low attachment surface	Provides an environment similar to in vivo conditions	Difficulty in mass production Lack of uniformity between spheroids	[27,28,29,30,31]

**Table 2 ijms-22-02491-t002:** Synthetic and natural hydrogels for 3D cell culture.

	Properties	Materials	Cells	Applications
Synthetic	Provide structural support to various cell types	PVA	Mouse 129 teratocarcinoma AT805 derived cells (ATDC5) [37], Human iPS cells (HPS0077) [38]	Repair cartilage [37], promote differentiation [38]
		pHEMA	Bovine ear chondrocytes [39]	Proliferate chondrocytes [39]
		PEG	Ovarian Follicle cell [40], human mesenchymal stem cells (hMSCs) [41]	Promote cell survival, growth [40], and viability by encapsulation [41]
Natural	Support cellular activities and are biocompatible and biodegradable	Collagen	Human umbilical vein endothelial cells (HUVECs) [42]	Form stable EC networks [42]
		Alginate	Human adipose-derived stem cells (hASCs) [43], rat astroglioma (LRM55) [44]	Maintain their ability to secrete therapeutic factors [43], maintain the viability and function [44]
		Hyaluronic acid	Human induced pluripotent stem cell-derived neural progenitor cells (hiPSC-NPCs) [45], human breast cancer MCF-7 cells [46]	Promote neural differentiation [45], higher tumorigenic capability of MCF-7 cells [46]

**Table 3 ijms-22-02491-t003:** Fabrication of porous scaffolds: advantages and disadvantages.

Method	Advantages	Disadvantages	References
Particulate Leaching	Modulate pore size and porosity	Limited pore shape and size	[15]
Solvent Casting	Modulate pore size and porosity Easy incorporation of drugs within the scaffold	Low pore interconnectivity	[100,101]
Emulsion Templating	Modulate particle size, high porosity, interconnectivity	Difficulty in obtaining emulsions with sufficient monodispersity for crystallization	[16,102,103]
Gas Foaming	Modulate pore size and porosity Free of toxic organic solvents	Unexpected pore interconnectivity	[104,105,106]
Melt Molding	Modulate pore size and porosity	High temperature required when molding	[107]

**Table 4 ijms-22-02491-t004:** Fabrication of fibrous scaffolds: advantages and disadvantages.

Method	Advantages	Disadvantages	References
Fiber Mesh	High surface area for cell attachment	Low structural stability	[21]
Fiber Bonding	High surface to volume ratio, high porosity	Limited applications to other polymers	[132]
Electrospinning	Induces cell alignment and directionality	Limited by cell seeding	[143,144,145]
Phase Separation	No reduction in the activity of molecules	Difficult to control the scaffold morphology	[138,146]
Self-Assembly	Form extremely stable scaffolds, less use of organic solvent	Expensive material, complicated and elaborate process	[147,148]

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
