# Peer review of "Applications of Biomaterials in 3D Cell Culture and Contributions of 3D Cell Culture to Drug Development and Basic Biomedical Research"

_ijms, 2021, doi:10.3390/ijms22052491_

Round 1

Reviewer 1 Report

Review on ijms-1063757

Title of the manuscript: Three-Dimensional Cell Culture Using Biomaterials and Applications

By YuJin Park, Kang Moo Huh,and Sun-Woong Kang

The manuscript is a review article on the applications of 3D cell culture in vitro. While the aim of the study is not clearly stated, the authors indicate that they chose to focus on the use of biomaterials for 3D cell culture and three areas of applications, such as cancer research, stem cells studies and drug development/toxicity screening.

This is generally accepted that 3D cell culture represents a major advancement in cell biology, biomedicine and tissue engineering. Its’ value as an animal-free preclinical testbed or disease model of a level up vs conventional monolayer cell cultures is also widely recognised. Numerous reviews on the subject have been published already. Considering this, the topic of the current review looks interesting and useful but challenging. In light of the available literature, it would be reasonable to expect seeing a mature, thought-provoking or visionary review under the title mentioned above.

However, and very unfortunately, the presented manuscript has a limited scope, shallow depth and rather listing a random fact than providing the analysis of the current state of the field.

First, the overview of the biomaterials used for 3D cell culture is significantly incomplete and, sometimes, it’s misleading. As the authors mention at the beginning that biomaterials in 3D cell culture are a kind of surrogate or replacement of the natural extracellular matrix of the tissues, the inclusion of the scaffold-free systems in the review (e.g., spheroids, organoids and embryoid bodies) along with a particular focus on this methodology is confusing. Also, due to the same reasons, the analysis and discussion of the ultra-low adhesion surfaces in the context of the declared topic are excessive and misleading.

In the review of synthetic and natural polymers as scaffolding materials, only a very superficial presentation on the subject is given. Importantly, there are no signs of analysis of state-of-the-art materials and new ideas in the field. In contrast, there is mostly a repetitive listing of common pieces of knowledge. Next, some statements are not corresponding to the cited literature. For example, in rows 274-275, the authors cite [134] and indicate on the usage of the cited protocol for creation of organoids and drug testing. However, the cited paper describes a methodology of whole-organ perfusion decellularisation and repopulation of whole decellularised scaffolds with hepatocytes and endothelial cells for the reconstruction of a vascularised bioengineered liver (for regenerative medicine applications). At the same time, the authors did not consider or mention the decellularised native tissue as biomaterials for 3D cell culture.

The same sad truth (limited depth, limited scope, no vision, no ideas) can be said about the analysis of the applications of 3D culture given in this manuscript.

Finally, the English language aspect of the paper needs significant improvement. The grammar is far from perfect; there are many repeating words in the neighbouring phrases, and not completely explainable usage of single forms of the nouns instead of plural ones in some places. Also, all the used abbreviations should be appropriately introduced.

Regrettably, I am recommending the rejection of the manuscript in the current state as its’ quality if not satisfying the criteria of Q1 journal publication.

Author Response

Response to reviewer (International Journal of Molecular Sciences)

Manuscript ID: ijms-1063757

Manuscript Title: Three-Dimensional Cell Culture Using Biomaterials and Applications

The authors thank the reviewers for thoughtful comments.

Reviewer 1:

The manuscript is a review article on the applications of 3D cell culture in vitro. While the aim of the study is not clearly stated, the authors indicate that they chose to focus on the use of biomaterials for 3D cell culture and three areas of applications, such as cancer research, stem cells studies and drug development/toxicity screening.

This is generally accepted that 3D cell culture represents a major advancement in cell biology, biomedicine and tissue engineering. Its’ value as an animal-free preclinical testbed or disease model of a level up vs conventional monolayer cell cultures is also widely recognised. Numerous reviews on the subject have been published already. Considering this, the topic of the current review looks interesting and useful but challenging. In light of the available literature, it would be reasonable to expect seeing a mature, thought-provoking or visionary review under the title mentioned above.

However, and very unfortunately, the presented manuscript has a limited scope, shallow depth and rather listing a random fact than providing the analysis of the current state of the field.

First, the overview of the biomaterials used for 3D cell culture is significantly incomplete and, sometimes, it’s misleading. As the authors mention at the beginning that biomaterials in 3D cell culture are a kind of surrogate or replacement of the natural extracellular matrix of the tissues, the inclusion of the scaffold-free systems in the review (e.g., spheroids, organoids and embryoid bodies) along with a particular focus on this methodology is confusing. Also, due to the same reasons, the analysis and discussion of the ultra-low adhesion surfaces in the context of the declared topic are excessive and misleading.

In the review of synthetic and natural polymers as scaffolding materials, only a very superficial presentation on the subject is given. Importantly, there are no signs of analysis of state-of-the-art materials and new ideas in the field. In contrast, there is mostly a repetitive listing of common pieces of knowledge. Next, some statements are not corresponding to the cited literature. For example, in rows 274-275, the authors cite [134] and indicate on the usage of the cited protocol for creation of organoids and drug testing. However, the cited paper describes a methodology of whole-organ perfusion decellularisation and repopulation of whole decellularised scaffolds with hepatocytes and endothelial cells for the reconstruction of a vascularised bioengineered liver (for regenerative medicine applications). At the same time, the authors did not consider or mention the decellularised native tissue as biomaterials for 3D cell culture.

The same sad truth (limited depth, limited scope, no vision, no ideas) can be said about the analysis of the applications of 3D culture given in this manuscript.

Finally, the English language aspect of the paper needs significant improvement. The grammar is far from perfect; there are many repeating words in the neighbouring phrases, and not completely explainable usage of single forms of the nouns instead of plural ones in some places. Also, all the used abbreviations should be appropriately introduced.

Regrettably, I am recommending the rejection of the manuscript in the current state as its’ quality if not satisfying the criteria of Q1 journal publication.

1. The aim of the study is not clearly stated.

Answer: As the reviewer commented, we have now added scientific contents in Introduction section for clear aim of study. And, we have changed title of manuscript to “Applications of biomaterials in 3D cell culture and contribution of 3D cell culture”

2. the presented manuscript has a limited scope, shallow depth and rather listing a random fact than providing the analysis of the current state of the field. First, the overview of the biomaterials used for 3D cell culture is significantly incomplete and, sometimes, it’s misleading.

Answer: As the reviewer commented, we have now classified the types of biomaterials for 3D cell culture into four types: hydrogel, solid scaffold, decellularized native tissue, and attachment resistant cell surface, and added examples of application as recent studies.

3. Some statements are not corresponding to the cited literature. At the same time, the authors did not consider or mention the decellularised native tissue as biomaterials for 3D cell culture.

Answer: As the reviewer commented, we have now carefully revised the references or added them as the latest data for throughout the manuscript. And, we have added the decellularized native tissue as biomaterials for 3D cell culture.

4. The same sad truth (limited depth, limited scope, no vision, no ideas) can be said about the analysis of the applications of 3D culture given in this manuscript.

Answer: As the reviewer commented, we have now revised contributions of 3D cell culture including cancer research and stem cell research.

5. The English language aspect of the paper needs significant improvement.

Answer: As the reviewer commented, we have now received English correctios to ensure any errors in spelling, grammar, and word choice from native speakers in American Journal Experts.

Reviewer 2 Report

The present review describes several biomaterials that are currently used in the field as crucial tools for achieving 3D cell culture systems. While the authors discuss various biomaterial scaffolds in great detail, little attention was given to the specific cell types that are able to be cultured on each scaffold. The authors mention essential factors to consider while choosing biomaterials, but do not give the respective details on what scaffolds offer greater success in recapitulating in vivo phenotypes. With these additions, this review will be acceptable for publication.

  •  Introduction
    • Random mention of chondrocytes in line 29 that doesn’t add anything
      • 2D culture systems section does not mention transwell membranes
      • Surely other cells would offer better examples?
    • Lines 38-40 what types of cells?
    • Lines 39-40 what cells?
    • Line 45 Define ULA surface.
  1. Biomaterials to improve cellular function
    • Lines 63-65 Describe cells from specific example and how alginate improved the cell culture.
    • When describing synthetic vs natural hydrogels, a table would be a great addition to compare hydrogels and determine which is best for specific cell types.
    • Overall flow of section is lacking. It reads like a list of hydrogels with little-to-no transitions across paragraphs.
    • Lines 93-94: Does this activity and therapeutic efficacy observed with cells cultured on HA correlate with what is seen in vivo? If not, then this is not relevant to mimic cellular environment.
    • Collagen, HA, Alginate were all mentioned in the hydrogels section… Is there another way to organize/structure the biomaterials section? Since natural hydrogels are included in solid scaffolds, it might be better to organize the review by rearranging the biomaterials section based on scaffold. For example, split into solid scaffolds, porous scaffolds, fibrous scaffolds, and scaffold-free systems.
    • Lines 178-182: I like the info on relating spheroid models to in vivo. This would be nice to include for other examples as well.
  1. Applications of 3D cell culture
    • Lines 194-196: Add references showing discrepancies between in vitro and clinical data.
    • Lines 219-222: Add references and more detail on 3D co-culture of cancer and non-cancerous cells.
    • Lines 244-246: Add references to studies being broadly discussed.
    • Lines 256-262: While organoids are very important for mimicking in vivo outcomes, it is a little confusing why the authors are discussing microfluidic devices here. Are biomaterials used in cell cultures plated on multielectrode arrays? If so, please include this information in this section.
    • Regarding the drug and toxicity screening section, this is included in all of the applications sections. It seems like this section can be removed and simply discussed further in the cancer cell culture and stem cell sections. Also, this section does not add much when organoid culture platforms are not used high-throughput. Adding information on specific known screening platforms with 3D culture systems would be beneficial in this section.

Author Response

Response to reviewer (International Journal of Molecular Sciences)

Manuscript ID: ijms-1063757

Manuscript Title: Three-Dimensional Cell Culture Using Biomaterials and Applications

The authors thank the reviewers for thoughtful comments.

Reviewer 2:

The present review describes several biomaterials that are currently used in the field as crucial tools for achieving 3D cell culture systems. While the authors discuss various biomaterial scaffolds in great detail, little attention was given to the specific cell types that are able to be cultured on each scaffold. The authors mention essential factors to consider while choosing biomaterials, but do not give the respective details on what scaffolds offer greater success in recapitulating in vivo phenotypes. With these additions, this review will be acceptable for publication.

Answer: As the reviewer commented, we have now added information of specific cell types that are cultured on each scaffold for 3D cell culture. In addition, for 3D cell culture, the advantages of each biomaterials and example of use have been added in the manuscript.

Introduction

Random mention of chondrocytes in line 29 that doesn’t add anything.

Answer: As the reviewer commented, we have now added information of chondrocytes.

2D culture systems section does not mention transwell membranes.

Answer: As the reviewer commented, we have now added 2D culture system using transwell membranes.

Surely other cells would offer better examples?

Answer: As the reviewer commented, we have now added primary human hepatocytes as better example.

Lines 38-40 what types of cells?

Answer: The cells presented in line 38-40 are various types of cancer cell lines and mesenchymal stem cells.

Line 45 Define ULA surface.

Answer: We have now changed “ULA surface” to “attachment resistant cell surface”.

Biomaterials to improve cellular function

1. Lines 63-65 Describe cells from specific example and how alginate improved the cell culture.

Answer: As the reviewer commented, we have now added information of the advantages of alginate and example of 3D culture have been added in the manuscript.

2. When describing synthetic vs natural hydrogels, a table would be a great addition to compare hydrogels and determine which is best for specific cell types.

Answer: As the reviewer commented, we have now added revised table 2 to compare hydrogels and determine which is best for specific cell types.

3. Overall flow of section is lacking. It reads like a list of hydrogels with little-to-no transitions across paragraphs.

Answer: As the reviewer commented, we have now reorganized the overall flow and added content around the latest research trends.

4. Lines 93-94: Does this activity and therapeutic efficacy observed with cells cultured on HA correlate with what is seen in vivo? If not, then this is not relevant to mimic cellular environment.

Answer: As the reviewer commented, we have now revised the sentence to make it easier to understand. “Other researchers have reported that when cells are 3D cultured in the presence of HA, the activity of multidrug resistance proteins is enhanced and therapeutic effectiveness is reduced compared to the 2D cultured cells”

5. Collagen, HA, Alginate were all mentioned in the hydrogels section… Is there another way to organize/structure the biomaterials section? Since natural hydrogels are included in solid scaffolds, it might be better to organize the review by rearranging the biomaterials section based on scaffold. For example, split into solid scaffolds, porous scaffolds, fibrous scaffolds, and scaffold-free systems.

Answer: As the reviewer commented, we have now reorganized the overall flow and added content around the latest research trends. Applications of biomaterials used for 3D cell culture were divided into hydrogel, solid scaffold, decellularized native tissue, and attachment resistant cell surface, and hydrogels were divided into synthetic and natural polymer.

6. Lines 178-182: I like the info on relating spheroid models to in vivo. This would be nice to include for other examples as well.

Answer: As the reviewer commented, we have now added other examples for spheroids.

Applications of 3D cell culture

7. Lines 194-196: Add references showing discrepancies between in vitro and clinical data.

Answer: As the reviewer commented, we have now added references showing discrepancies between in vitro and clinical data.

8. Lines 219-222: Add references and more detail on 3D co-culture of cancer and non-cancerous cells.

Answer: As the reviewer commented, we have now added references showing 3D co-culture of cancer and non-cancerous cells.

9. Lines 244-246: Add references to studies being broadly discussed.

Answer: As the reviewer commented, we have now added references to studies being broadly discussed.

10. Lines 256-262: While organoids are very important for mimicking in vivo outcomes, it is a little confusing why the authors are discussing microfluidic devices here. Are biomaterials used in cell cultures plated on multielectrode arrays? If so, please include this information in this section.

Answer: We appreciate the reviewer’s comment and agree with the reviewer in that microfluidic devices are confusing here. Thus, we have now changed the subtitle from “stem cell research” to “stem cell research and drug screening”. In addition, we have added organoids as model for drug screening. And, microfluidic devices can be able to realize analysis of functions of various organoids for drug screening.

11. Regarding the drug and toxicity screening section, this is included in all of the applications sections. It seems like this section can be removed and simply discussed further in the cancer cell culture and stem cell sections. Also, this section does not add much when organoid culture platforms are not used high-throughput. Adding information on specific known screening platforms with 3D culture systems would be beneficial in this section.

Answer: As the reviewer commented, we have now removed drug screening section and included drug screening in the cancer research and stem cell research section.

Reviewer 3 Report

Dear editors:  

 It is a great honor and pleasure for me to be invited as the reviewer for this important work. Park et al. comprehensively reviewed the current biomaterials used for three-dimensional (3D) cell culture to improve cellular functions along with the application of 3D cell culture in simulating the physical and biological functions of tissues in human body. This study topic is interesting and novel, attributing to Prof. Huh’s and Prof. Kang’s long-term efforts and contributions in this scientific field. The review work is comprehensive. I have a number of comments concerning this study:

  1. Since the term “three-dimensional (3D)” was expressed in the abstract, the full names of 3D and 2D should be used in the text for the first time.
  2. Line 45: The full names of ULA should be spelled out here and also in legends of Figure 1.
  3. The title of Figure 1 could be rephrased as “Types of biomaterials used in 3D cell culture.” Extensive editing of English language and style are still required.
  4. Line 45: “,” is missing before the word “including”. The language would benefit from some language polishing; articles (a, the) are missing in some places, for example, “The” knowledge of 3D culture methods has significantly increased, resulting in the development of numerous applications.
  5. Line 67: The full names of “PVA, pHEMA, PEG” should be used in the text for the first time, also in Line 312
  6. Line 85: “escherichia coli” should be expressed in italic font.
  7. Line 104-105: The full names of “hESCs, hiPSCs” should be spelled out in the text for the first time.
  8. Line 178-180: “in vivo” should be expressed in italic font.
  9. Line 274: “in vitro” should be expressed in italic font.
  10. Line 312: All the abbreviations in the text should be presented here.

Thank you for giving me the opportunity to review this interesting article. After minor revision, this important review article should be published as soon as possible.

Author Response

Response to reviewer (International Journal of Molecular Sciences)

Manuscript ID: ijms-1063757

Manuscript Title: Three-Dimensional Cell Culture Using Biomaterials and Applications

The authors thank the reviewers for thoughtful comments.

Reviewer 3:

It is a great honor and pleasure for me to be invited as the reviewer for this important work. Park et al. comprehensively reviewed the current biomaterials used for three-dimensional (3D) cell culture to improve cellular functions along with the application of 3D cell culture in simulating the physical and biological functions of tissues in human body. This study topic is interesting and novel, attributing to Prof. Huh’s and Prof. Kang’s long-term efforts and contributions in this scientific field. The review work is comprehensive. I have a number of comments concerning this study:

  1. Since the term “three-dimensional (3D)” was expressed in the abstract, the full names of 3D and 2D should be used in the text for the first time.

Answer: As the reviewer commented, we have now added “three-dimensional (3D)” and “two-dimensional (2D)” in the text for the first time.

  1. Line 45: The full names of ULA should be spelled out here and also in legends of Figure 1.

Answer: As the reviewer commented, we have now changed “ULA surface” to “attachment resistant cell surface”

  1. The title of Figure 1 could be rephrased as “Types of biomaterials used in 3D cell culture.” Extensive editing of English language and style are still required.

Answer: As the reviewer commented, we have now changed the title of figure 1 as “Types of biomaterials used in 3D cell culture”. And, we have now received English correctios to ensure any errors in spelling, grammar, and word choice from native speakers in American Journal Experts.

  1. Line 45: “,” is missing before the word “including”. The language would benefit from some language polishing; articles (a, the) are missing in some places, for example, “The” knowledge of 3D culture methods has significantly increased, resulting in the development of numerous applications.

Answer: As the reviewer commented, we have now received English correctios to ensure any errors in spelling, grammar, and word choice from native speakers in American Journal Experts.

  1. Line 67: The full names of “PVA, pHEMA, PEG” should be used in the text for the first time, also in Line 312

Answer: As the reviewer commented, we have now added “Poly vinyl alcohol (PVA), Poly-2-hydroxyethyl methacrylate (pHEMA) and Poly ethylene glycol (PEG)” in the text for the first time.

  1. Line 85: “escherichia coli” should be expressed in italic font.

Answer: As the reviewer commented, we have now changed to “escherichia coli” in italic font.

  1. Line 104-105: The full names of “hESCs, hiPSCs” should be spelled out in the text for the first time.

Answer: As the reviewer commented, we have now added “embryonic stem cells (ESCs) and induced pluripotent stem cells (iPSCs)” in the text for the first time.

  1. Line 178-180: “in vivo” should be expressed in italic font.

Answer: As the reviewer commented, we have now changed to “in vivo” in italic font.

  1. Line 274: “in vitro” should be expressed in italic font.

Answer: As the reviewer commented, we have now changed to “in vitro” in italic font.

  1. Line 312: All the abbreviations in the text should be presented here.

Answer: As the reviewer commented, we have now added all the abbreviations in the text.

Reviewer 4 Report

This manuscript introduces some types of biomaterials and the application fields in three-dimensional cell culture technology. The subject is novel and useful. However, there are some problems with this manuscript. If this manuscript is accepted for publication in the journal, the author should do a major revise. 

  1. The title of this article covers a wide range of biomaterials,but this manuscript only introduced hydrogel and solid scaffold in detail. The author has given us a detailed classification in Table 1, and the author should provide us with supplemental statements of other biomaterials. In Fig 2, the author describes the proliferation and differentiation of stem cells, but the author does not show the process of cells' three-dimensional culture. 
  2. In recent years, various experimental 3D cell culture models have been developed, and it is better to add other new areas and technology of the three-dimensional cell culture in this manuscript. 
  3. The content of biomaterials used in the 3D cell culture introduced in this paper is not enough. The author should add some details about the relationship between 3D cell culture and biomaterials, such as selecting biomaterials in 3D culture and various biomaterials' characteristics. Moreover, the author should give more information about 3D cell culture's progress in bone tissue engineering research and clinical application. 
  4. The author should pay attention to the grammar mistakes. In line 18," for improving of cellular functions," the "of" should be deleted. Some paper could be cited in this paper:BMC Biomedical Engineering 2021 3:1/BMC Biomedical Engineering 2020 2:11/ BMC Biomedical Engineering 2020 2:10/ BMC Biomedical Engineering 2020 2:4/Research. 2020; 4907185. 
  5. In line 181, "oxygen and nutrients is not well penetrated to the inside of the spheroid," "is" should be replaced by "are." 
  6. In line 92,"…. by enhanced….", the verb form is an error. 

Author Response

Response to reviewer (International Journal of Molecular Sciences)

Manuscript ID: ijms-1063757

Manuscript Title: Three-Dimensional Cell Culture Using Biomaterials and Applications

The authors thank the reviewers for thoughtful comments.

Reviewer 4:

This manuscript introduces some types of biomaterials and the application fields in three-dimensional cell culture technology. The subject is novel and useful. However, there are some problems with this manuscript. If this manuscript is accepted for publication in the journal, the author should do a major revise.

1. The title of this article covers a wide range of biomaterials,but this manuscript only introduced hydrogel and solid scaffold in detail. The author has given us a detailed classification in Table 1, and the author should provide us with supplemental statements of other biomaterials. In Fig 2, the author describes the proliferation and differentiation of stem cells, but the author does not show the process of cells' three-dimensional culture.

Answer: We appreciate the reviewer’s comment and agree with the reviewer. We have now added scientific contents in Introduction section for clear aim of study. In addition, we have now changed title of manuscript to “Applications of biomaterials in 3D cell culture and contribution of 3D cell culture”. And, we have revised Table 1, Table 2, Figure 1. Finally, we have added the process of 3D cell culture in Figure 1.

2. In recent years, various experimental 3D cell culture models have been developed, and it is better to add other new areas and technology of the three-dimensional cell culture in this manuscript.

Answer: As the reviewer commented, we have now added new areas and technology of 3D cell culture in the manuscript and replaced 89 references from recent studies since 2018.

3. The content of biomaterials used in the 3D cell culture introduced in this paper is not enough. The author should add some details about the relationship between 3D cell culture and biomaterials, such as selecting biomaterials in 3D culture and various biomaterials' characteristics. Moreover, the author should give more information about 3D cell culture's progress in bone tissue engineering research and clinical application.

Answer: As the reviewer commented, we have now added the content of biomaterials used in the 3D cell culture and bone tissue engineering research and clinical applications.

4. The author should pay attention to the grammar mistakes. In line 18," for improving of cellular functions," the "of" should be deleted. Some paper could be cited in this paper:BMC Biomedical Engineering 2021 3:1/BMC Biomedical Engineering 2020 2:11/ BMC Biomedical Engineering 2020 2:10/ BMC Biomedical Engineering 2020 2:4/Research. 2020; 4907185.

Answer: As the reviewer commented, we have now received English correctios to ensure any errors in spelling, grammar, and word choice from native speakers in American Journal Experts. And, we have added some of the recommended references.

5. In line 181, "oxygen and nutrients is not well penetrated to the inside of the spheroid," "is" should be replaced by "are."

Answer: As the reviewer commented, we have now received English correctios to ensure any errors in spelling, grammar, and word choice from native speakers in American Journal Experts.

6. In line 92,"…. by enhanced….", the verb form is an error.

Answer: As the reviewer commented, we have now received English correctios to ensure any errors in spelling, grammar, and word choice from native speakers in American Journal Experts.

Reviewer 5 Report

The review by Park et al. entitled "Three-Dimensional Cell Culture Using Biomaterials and Applications" generally lacks novelty, uniformity, and structure. In this form, I do not believe that it is suitable for publishing.

The introduction must be improved as it does not clearly specify and introduce the main ideas which will be discussed throughout the manuscript.

The first chapter categorizes 3D cell cultures in hydrogels, solid scaffolds, and ULA systems. However, hydrogels can also be considered as solid scaffolds.  Additionally, the chapter does not even describe all 3D culture system types mentioned in Table 1. Therefore, the chapter should be entitled Types of 3D cell culture systems and should include all the systems within Table 1.

Moreover, hydrogels are further divided into synthetic and natural hydrogels, but only natural hydrogels are described and exemplified. But there are no novel biomaterials presented for such applications, as collagen, hyaluronic acid, and alginate are highly known and utilized in the literature.

The subchapter on solid scaffolds briefly describes only three types, without bringing any knowledge to the field, as the information presented is rather old.

The ULA system subchapter is not sufficiently described and explained.

The application chapter is too short. There are only two or three studies described and the manuscript does not bring any significance to the research field.

The subchapter on Drug and toxicity screening is irrelevant, as there is only paragraph that does not describe anything.

English must be significantly corrected and improved.

Only 34 references are new, published after 2018, which limits the novelty of the manuscript.

Minor revisions

  • References should be inserted as [1].
  • Table 1 - the classification is not very relevant and rather has some errors, e.g., hydrogels are also solid scaffolds, micropatterned surfaces are not scaffolds, etc.
  • Line 45 - what does ULA system mean? You must explain.
  • Figure 1 - images within the figure seem to be taken from other sources, which are not quoted.

Author Response

Response to reviewer (International Journal of Molecular Sciences)

Manuscript ID: ijms-1063757

Manuscript Title: Three-Dimensional Cell Culture Using Biomaterials and Applications

The authors thank the reviewers for thoughtful comments.

Reviewer 5:

The review by Park et al. entitled "Three-Dimensional Cell Culture Using Biomaterials and Applications" generally lacks novelty, uniformity, and structure. In this form, I do not believe that it is suitable for publishing.

1. The introduction must be improved as it does not clearly specify and introduce the main ideas which will be discussed throughout the manuscript.

Answer: As the reviewer commented, we have now added scientific contents in Introduction section for clear aim of study. In addition, we have changed title of manuscript to “Applications of biomaterials in 3D cell culture and contribution of 3D cell culture”. And, we also added the latest content on 3D cell culture throughout the manuscript.

2. The first chapter categorizes 3D cell cultures in hydrogels, solid scaffolds, and ULA systems. However, hydrogels can also be considered as solid scaffolds. Additionally, the chapter does not even describe all 3D culture system types mentioned in Table 1. Therefore, the chapter should be entitled Types of 3D cell culture systems and should include all the systems within Table 1.

Answer: As the reviewer commented, we have now classified the types of biomaterials for 3D cell culture into four types: hydrogel, solid scaffold, decellularized native tissue, and attachment resistant cell surface, and added examples of application as recent studies throughout the manuscript. And, we have revised Table 1, Table 2, Figure 1.

3. Moreover, hydrogels are further divided into synthetic and natural hydrogels, but only natural hydrogels are described and exemplified. But there are no novel biomaterials presented for such applications, as collagen, hyaluronic acid, and alginate are highly known and utilized in the literature.

Answer: As the reviewer commented, we have now added scientific contents for hydrogels.

4. The subchapter on solid scaffolds briefly describes only three types, without bringing any knowledge to the field, as the information presented is rather old.

Answer: As the reviewer commented, we have now added information of specific cell types that are cultured on each scaffold for 3D cell culture. In addition, for 3D cell culture, the advantages of each biomaterials and example of use have been added in the manuscript.

5. The ULA system subchapter is not sufficiently described and explained.

Answer: We have now changed “ULA surface” to “attachment resistant cell surface” and added scientific contents for attachment resistant cell surface.

6. The application chapter is too short. There are only two or three studies described and the manuscript does not bring any significance to the research field.

Answer: As the reviewer commented, we have now added scientific contents in application chapter.

7. The subchapter on Drug and toxicity screening is irrelevant, as there is only paragraph that does not describe anything.

Answer: As the reviewer commented, we have now deleted drug and toxicity screening section. And “drug and toxicity screening” was added to “cancer research” and “stem cell research” section.

8. English must be significantly corrected and improved.

Answer: As the reviewer commented, we have now received English correctios to ensure any errors in spelling, grammar, and word choice from native speakers in American Journal Experts.

9. Only 34 references are new, published after 2018, which limits the novelty of the manuscript.

Answer: As the reviewer commented, we have now added new areas and technology of 3D cell culture in the manuscript and replaced 89 references from recent studies since 2018.

Minor revisions

10. References should be inserted as [1].

Answer: As the reviewer commented, we have now revised references as [1].

11. Table 1 - the classification is not very relevant and rather has some errors, e.g., hydrogels are also solid scaffolds, micropatterned surfaces are not scaffolds, etc.

Answer: As the reviewer commented, we have now revised table 1 and added scientific contents in the manuscript.

12. Line 45 - what does ULA system mean? You must explain.

Answer: As the reviewer commented, we have now changed “ULA surface” to “attachment resistant cell surface” and added scientific contents for attachment resistant cell surface.

13. Figure 1 - images within the figure seem to be taken from other sources, which are not quoted.

Answer: As the reviewer commented, we drew pictures for figure 1 ourselves.

Round 2

Reviewer 1 Report

Dear authors, thank you for your efforts to upgrade the manuscript. Indeed, a significant improvement has been made. However, unfortunately, the English language (grammar, correctness and scientific meaning of the phrases) requires more work. Also, the formatting of the manuscript is not consistent across the document. I started with a few corrections, but gave up as there are too many imperfections impairing the readability and blurring the message of the review. 

I am attaching the first few corrections that I recommend to consider and ask authors to pass through their manuscript further with the same level of attention.

I appreciate the work that has been done by the authors to upgrade the manuscript. This resulted in a significant improvement of the paper. I think the review is now almost ready to be published. However, I would recommend authors to address the following concerns before further editorial processing.

  1. The title of the manuscript “Applications of Biomaterials in 3D Cell Culture and Contributions of 3D Cell Culture” has some uncertainty. At least, grammatically, it is expected that the “contributions of 3D cell culture” will be the ”contributions to (something)”. Considering the paper/abstract context, it may be the contribution of 3D cell culture to drug development and basic biomedical research.
  2. Row 34 - “is not duplicated” - should be is “in not accurately reproduced”.
  3. Row 39- 40 : it’s unclear, why the authors compare the 2d culture with transwell culture, and why it is an improvement vs. 2d systems? Transwell is a kind of “layered” 2D system.
  4. Row 53 – please, remove the dot at the end of the phrase “Physical strength is easily adjusted.”
  5. Row 53 - “Leads to the loss of mechanical properties of the tissues as compared to the native group” – please, correct to a specific change (increase/decrease?) of specific mechanical property (stiffness? Roughnesss? Porosity? Elasticity? Tension strength? – etc.). “Loss of mechanical properties” sounds confusing and non-informative.
  6. Row 53 – for the attachment resistant surfaces – “Provides a similar physicochemical environment in vivo by facilitating cell-cell interactions” – please, correct the grammar (an environment similar to in vivo conditions). It also not facilitating the cell-cell interactions, but forcing it. It’s hardly likely that this actually represents a biologically accurate model of in vivo tissue. The limited spheroids’ size (again, please, correct the grammar) is not a single disadvantage of the ULA-based cultures.
  7. I am not sure why a new term (“attachment resistant surfaces”) is introduced. ULA is a known and widely used term.
  8. Row 55 – please, reconcile the text and the refs to the Table and Figure. The text says: “There are a number of 3D culture systems that are already available (Table 1). Biomaterials are also available in 3D cell culture to improve the efficiency of culture and cell functions in various forms, including hydrogels, solid scaffolds, decellularized native tissue, and attachment resistant cell 57 surface (Figure 1).” – At the same time the title of the Table 1 is “Types of biomaterials used in 3D cell culture” (instead of the types of 3D cell culture systems) and the figure 1 caption is very similar “Figure 1. Types of biomaterials used in 3D culture. A) Hydrogel, B) Solid scaffold, C) Decellularized native tissue D) Attachment resistant cell surface”. It looks like the authors do not differentiate the 3d culture systems and biomaterials. The Figure 1 depicts not the types of biomaterials, but the biomaterials and related method of 3d culture preparation. Also, the grammar of the above mentioned phrase requires correction (“Biomaterials are also available in 3D cell culture to improve the efficiency of culture…”) – biomaterials available FOR 3d cell culture, not IN. Why “also”? etc.
  9. Row 59 – “Thus, this review paper seeks to address the applications of biomaterials in 3D cell culture and the contribution of 3D cell culture to cancer 60 research, stem cell research, and drug and toxicity screening.” – Please, change to “This review addresses the applications of biomaterials in 3D cell culture and the contribution of 3D cell culture to cancer and stem cells research, and drug and toxicity screening.”
  10. R67-68, please, correct the grammar (polymeric materials … composed of polymers).
  11. R68-69 - These hydrogels can mimic … environments similar to the physiological surroundings of cells in the natural … environment (correct grammar and contents) – what are “physiological surroundings”? what are the “natural environment” – and what is the difference between them?..

Overall recommendation: I recommend this paper to publication after extensive correction of English language and style.

Thank you.

Author Response

Response to reviewer (International Journal of Molecular Sciences)

Manuscript ID: ijms-1063757

Manuscript Title: Three-Dimensional Cell Culture Using Biomaterials and Applications

The authors thank the reviewers for thoughtful comments.

Reviewer 1:

Dear authors, thank you for your efforts to upgrade the manuscript. Indeed, a significant improvement has been made. However, unfortunately, the English language (grammar, correctness and scientific meaning of the phrases) requires more work. Also, the formatting of the manuscript is not consistent across the document. I started with a few corrections, but gave up as there are too many imperfections impairing the readability and blurring the message of the review. I started with a few corrections, but gave up as there are too many imperfections impairing the readability and blurring the message of the review. I am attaching the first few corrections that I recommend to consider and ask authors to pass through their manuscript further with the same level of attention. I appreciate the work that has been done by the authors to upgrade the manuscript. This resulted in a significant improvement of the paper. I think the review is now almost ready to be published. However, I would recommend authors to address the following concerns before further editorial processing.

  1. The title of the manuscript “Applications of Biomaterials in 3D Cell Culture and Contributions of 3D Cell Culture” has some uncertainty. At least, grammatically, it is expected that the “contributions of 3D cell culture” will be the ”contributions to (something)”. Considering the paper/abstract context, it may be the contribution of 3D cell culture to drug development and basic biomedical research.

Answer: As the reviewer commented, we have now changed the title of manuscript to “Applications of biomaterials in 3D cell culture and contribution of 3D cell culture to drug development and basic biomedical research”.

  1. Row 34 - “is not duplicated” - should be is “in not accurately reproduced”.

Answer: As the reviewer commented, we have now changed “is not duplicated” to “is not accurately reproduced”.

  1. Row 39- 40 : it’s unclear, why the authors compare the 2d culture with transwell culture, and why it is an improvement vs. 2d systems? Transwell is a kind of “layered” 2D system.

Answer: As the reviewer commented, we have now changed to “Also, the transwell culture system, a kind of layered 2D culture system, was introduced as a co-culture system to simulate the in vivo environment”.

  1. Row 53 – please, remove the dot at the end of the phrase “Physical strength is easily adjusted.”

Answer: As the reviewer commented, we have now removed the dot at the end of the phrase “physical strength is easily adjusted”.

  1. Row 53 - “Leads to the loss of mechanical properties of the tissues as compared to the native group” – please, correct to a specific change (increase/decrease?) of specific mechanical property (stiffness? Roughnesss? Porosity? Elasticity? Tension strength? – etc.). “Loss of mechanical properties” sounds confusing and non-informative.

Answer: As the reviewer commented, we have now changed to “Decrease of mechanical properties (roughness, elasticity, and tension strength) of the tissues as compared to the native group”.

  1. Row 53 – for the attachment resistant surfaces – “Provides a similar physicochemical environment in vivo by facilitating cell-cell interactions” – please, correct the grammar (an environment similar to in vivo conditions). It also not facilitating the cell-cell interactions, but forcing it. It’s hardly likely that this actually represents a biologically accurate model of in vivo tissue. The limited spheroids’ size (again, please, correct the grammar) is not a single disadvantage of the ULA-based cultures.

Answer: As the reviewer commented, we have now changed to “provides an environment similar to in vivo conditions” and “Difficulty in mass production, Lack of uniformity between spheroids”.

  1. I am not sure why a new term (“attachment resistant surfaces”) is introduced. ULA is a known and widely used term.

Answer: As the reviewer commented, we have now changed to “Ultra-low attachment (ULA) surface”.

  1. Row 55 – please, reconcile the text and the refs to the Table and Figure. The text says: “There are a number of 3D culture systems that are already available (Table 1). Biomaterials are also available in 3D cell culture to improve the efficiency of culture and cell functions in various forms, including hydrogels, solid scaffolds, decellularized native tissue, and attachment resistant cell 57 surface (Figure 1).” – At the same time the title of the Table 1 is “Types of biomaterials used in 3D cell culture” (instead of the types of 3D cell culture systems) and the figure 1 caption is very similar “Figure 1. Types of biomaterials used in 3D culture. A) Hydrogel, B) Solid scaffold, C) Decellularized native tissue D) Attachment resistant cell surface”. It looks like the authors do not differentiate the 3d culture systems and biomaterials. The Figure 1 depicts not the types of biomaterials, but the biomaterials and related method of 3d culture preparation. Also, the grammar of the above mentioned phrase requires correction (“Biomaterials are also available in 3D cell culture to improve the efficiency of culture…”) – biomaterials available FOR 3d cell culture, not IN. Why “also”? etc.
  2. Answer: As the reviewer commented, we have now changed title of Table 1 and Figure 1. Also, we changed “in” to “for” in above mentioned phrase.

  1. Row 59 – “Thus, this review paper seeks to address the applications of biomaterials in 3D cell culture and the contribution of 3D cell culture to cancer 60 research, stem cell research, and drug and toxicity screening.” – Please, change to “This review addresses the applications of biomaterials in 3D cell culture and the contribution of 3D cell culture to cancer and stem cells research, and drug and toxicity screening.”
  2. Answer: As the reviewer commented, we have now changed to “This review addresses the applications of biomaterials in 3D cell culture and the contribution of 3D cell culture to cancer and stem cells research, and toxicity screening.”

  1. R67-68, please, correct the grammar (polymeric materials … composed of polymers).

Answer: As the reviewer commented, we have now changed to “Hydrogels have 3D structure, hydrophilic property, and polymeric networks capable of absorbing large amounts of water or biological fluids.”

  1. R68-69 - These hydrogels can mimic … environments similar to the physiological surroundings of cells in the natural … environment (correct grammar and contents) – what are “physiological surroundings”? what are the “natural environment” – and what is the difference between them?.
  2. Answer: As the reviewer commented, we have now changed to “These hydrogels can mimic soft and wet environments similar to ECM of native tissues….”

Overall recommendation: I recommend this paper to publication after extensive correction of English language and style.

Answer: As the reviewer commented, we have now corrected grammar of English language and style by native speaker.

Reviewer 4 Report

Good revision for publicaiton

Author Response

The authors thank the reviewers for thoughtful comments.

Reviewer 5 Report

The authors have addressed all my comments and suggestions and have significantly improved the quality of the manuscript. While it does not bring considerable novelty to the field, it is a recap of the most commonly used biomaterials for 3D cell cultures and their applications.

In this manner, I suggest the acceptance of the manuscript after some minor revisions (e.g., in Table 2, there should be a space before the reference).

Additionally, I suggest introducing the following references:

  • 10.1038/s41467-021-21029-2
  • 10.33263/briac96.474484
  • 10.2478/msp-2020-0054
  • 10.33263/briac96.439445
  • 10.4103/1673-5374.306097
  • 10.1002/adhm.201900670
  • 10.33263/briac96.452457

Author Response

Response to reviewer (International Journal of Molecular Sciences)

Manuscript ID: ijms-1063757

Manuscript Title: Three-Dimensional Cell Culture Using Biomaterials and Applications

The authors thank the reviewers for thoughtful comments.

Reviewer 5:

The authors have addressed all my comments and suggestions and have significantly improved the quality of the manuscript. While it does not bring considerable novelty to the field, it is a recap of the most commonly used biomaterials for 3D cell cultures and their applications.

In this manner, I suggest the acceptance of the manuscript after some minor revisions (e.g., in Table 2, there should be a space before the reference).

Answer: As the reviewer commented, we have now corrected Table 2.

Additionally, I suggest introducing the following references:

10.1038/s41467-021-21029-2

10.33263/briac96.474484

10.2478/msp-2020-0054

10.33263/briac96.439445

10.4103/1673-5374.306097

10.1002/adhm.201900670

10.33263/briac96.452457

Answer: As the reviewer commented, we have now added above mentioned references in the manuscript.